# Leveraging Immunofocusing and Virus-like Particle Display to Enhance Antibody Responses to the Malaria Blood-Stage Invasion Complex Antigen PfCyRPA

**DOI:** 10.3390/vaccines12080859

**Published:** 2024-07-30

**Authors:** Kasper H. Björnsson, Maria R. Bassi, Anne S. Knudsen, Kara-Lee Aves, Èlia Morella Roig, Adam F. Sander, Lea Barfod

**Affiliations:** 1Centre for translational Medicine and Parasitology, Department of Immunology and Microbiology (ISIM), Faculty of Health and Medical Sciences, University of Copenhagen, 2200 Copenhagen, Denmark; kasperhb@sund.ku.dk (K.H.B.); mbassi@sund.ku.dk (M.R.B.); asander@sund.ku.dk (A.F.S.); 2AdaptVac, Ole Maaløes Vej 3, 2200 Copenhagen, Denmark

**Keywords:** malaria, blood-stage vaccine development, PfCyRPA, immunofocusing, VLP, PCRCR invasion complex

## Abstract

A vaccine protecting against malaria caused by *Plasmodium falciparum* is urgently needed. The blood-stage invasion complex PCRCR consists of the five malarial proteins PfPTRAMP, PfCSS, PfRipr, PfCyRPA, and PfRH5. As each subcomponent represents an essential and highly conserved antigen, PCRCR is considered a promising vaccine target. Furthermore, antibodies targeting the complex can block red blood cell invasion by the malaria parasite. However, extremely high titers of neutralizing antibodies are needed for this invasion-blocking effect, and a vaccine based on soluble PfRH5 protein has proven insufficient in inducing a protective response in a clinical trial. Here, we present the results of two approaches to increase the neutralizing antibody titers: (A) immunofocusing and (B) increasing the immunogenicity of the antigen via multivalent display on capsid virus-like particles (cVLPs). The immunofocusing strategies included vaccinating with peptides capable of binding the invasion-blocking anti-PfCyRPA monoclonal antibody CyP1.9, as well as removing non-neutralizing epitopes of PfCyRPA through truncation. Vaccination with PfCyRPA coupled to the AP205 cVLP induced nearly two-fold higher IgG responses compared to vaccinating with soluble PfCyRPA protein. Immunofocusing using a linear peptide greatly increased the neutralizing capacity of the anti-PfCyRPA antibodies. However, significantly lower total anti-PfCyRPA titers were achieved using this strategy. Our results underline the potential of a cVLP-based malaria vaccine including full-length PfCyRPA, which could be combined with other leading malaria vaccine antigens presented on cVLPs.

## 1. Introduction

Malaria caused by the *Plasmodium falciparum* parasite remains one of the world’s largest health concerns despite efforts in drug distribution and mosquito countermeasures. An efficacious vaccine would be an indispensable weapon against the disease. A promising vaccine target is a pentameric invasion complex consisting of the proteins PfPTRAMP, PfCSS, PfRipr, PfCyRPA, and PfRH5 (PCRCR) [1,2,3,4,5,6,7]. The complex facilitates the malaria parasite’s invasion of the red blood cell (RBC). Merozoite invasion is a crucial step in the parasite’s life cycle, and blocking this process leads to parasite death. All five complex proteins have been shown to be essential for the parasite life cycle, as RBC invasion is prevented by conditional knockout of PfPTRAMP, PfCSS, PfRipr, and PfCyRPA [1,6] and by blocking PfRH5’s binding to basigin [8]. Further, the complex is highly conserved [1,9,10,11], which could lead to high strain-transcendent protection induced by a PCRCR-targeting vaccine and suggests functional restraints on possible mutations. Antibodies targeting PfRH5 [12,13], PfRipr [14,15], and PfCyRPA [1,4,16,17] have all shown strain-transcending, invasion-blocking capabilities. Nanobodies or monoclonal antibodies (mAb) with parasite-growth-inhibitory activity (GIA) have been identified for all five complex proteins [1,2,3,7,12,16,18,19,20], as well as polyclonal IgG targeting PfRh5, PfCyRPA, and PfRipr [2,3,4,21,22]. Polyclonal IgG purified from mice immunized with 20 µg of either PfRipr-, PfCyRPA-, or PfRH5-soluble protein showed similar in vitro GIA [23].

The vaccination of non-human primates with PfRH5 conferred protection in a challenge model [24], as did passive immunization with an anti-PfRH5 monoclonal antibody [25]. In humans, only one challenge study has been conducted, which used soluble PfRH5 in AS01 adjuvant for vaccinations. This study demonstrated the limited efficacy of the RH5-based vaccine, with the immunized group showing 10.5 days to diagnosis compared to 9.5 days in the control group, as well as a modest reduction in the parasite multiplication rate by roughly 20% [26].

PfCyRPA has also emerged as a promising immunogen, as it can induce antibodies capable of neutralizing *P. falciparum*’s invasion of RBCs in vitro [4] and can be expressed with high yields [27]. Furthermore, several neutralizing mAbs against the protein have been characterized, and the structures of five of these have been solved, thereby aiding in the guidance of vaccine design [16,18,19,28].

GIA assays [29] have been widely used to determine the neutralizing capabilities of antibodies targeting merozoite antigens [15,16,17,23,26,30]. The assay exhibits a positive correlation between IgG concentration and neutralization. Further, a correlation between in vitro and in vivo growth inhibition has been shown, rendering GIA a valuable proxy for in vivo efficacy of induced antibodies [26]. The inefficacy of the PfRh5-based vaccine could therefore partially be explained by the insufficient induction of neutralizing antibodies in the vaccinees compared to the concentrations used in the GIA assay to achieve high parasite invasion inhibition. Two potential strategies to address this issue include (A) immunofocusing by presenting mainly neutralizing epitopes and (B) enhancing the immunogenicity of the antigen by presenting it in a multivalent manner on capsid virus-like particles (cVLPs).

cVLPs have shown great potential in providing long-lasting, high-titer antibody responses. Their size provides efficient lymphatic drainage and dendritic cell uptake, and their high-density, repetitive antigen display enables efficient B-cell receptor crosslinking, conferring strong humoral responses [31,32,33,34]. The AP205 phage capsid protein self-assembles from 180 AP205 phage capsid subunits and has been used to develop a versatile cVLP-based vaccine platform employing a split-protein Tag/Catcher conjugation system [35] to facilitate high-density, unidirectional antigen display [36]. This Tag/Catcher-AP205 cVLP platform has exhibited significant enhancement in immunogenicity across various displayed antigens in numerous preclinical studies [36,37,38,39,40]. Notably, this technology was recently utilized in the development of a COVID-19 vaccine [37], which demonstrated robust safety and immunogenicity profiles throughout clinical phases I, II, and III (ClinicalTrials.gov identifiers: NCT04839146 [41], NCT05329220, and NCT05077267). Additionally, the Tag/Catcher-AP205 cVLP platform has been shown to maintain efficacy after freeze-drying and extended storage at room temperature, rendering it optimal for areas where maintaining a cold chain is challenging [42]. Previously, the AP205 cVLP platform has been used to display PfCyRPA, which yielded increased anti-PfCyRPA titers upon immunization compared to those induced by soluble PfCyRPA protein [38]. The changes in the RBC-invasion neutralizing capacity of the cVLP-induced antibodies, however, have not been investigated. 

Following the idea of Reverse Vaccinology 2.0 [43], it is possible that immunization with a construct that harbors the epitope of a characterized neutralizing antibody can elicit an antibody response with similar neutralizing capabilities. By removing the non-neutra-lizing epitopes from the immunogen, it may be possible to focus the immune response and therefore increase the titers of the desired neutralizing antibodies.

The epitopes of several neutralizing mAbs targeting PfCyRPA have been identified. PfCyRPA is a six-bladed, β-propeller sheet protein that couples together PfRH5 and PfRipr in the PCRCR complex. Three independent studies have solved the structures of a total of five different murine-neutralizing mAbs binding to PfCyRPA [18,19,28]. One mAb primarily binds blade two (Cy.007 [28]), two mAbs primarily bind blade three (8A7 [18] + Cy.003 [28]), and two mAbs bind both blade two and three (c12 [19] + Cy.004 [28]). Furthermore, a previous study from our group showed two neutralizing (CyP1.9 and CyP2.39) and one synergistic (CyP2.27) mAb to bind a truncation of PfCyRPA consisting of only blades two and three [16]. This truncation has been named Fragment A. Thus, Fragment A is an obvious candidate for a Reverse Vaccinology 2.0 approach. Identifying an even smaller fragment capable of binding neutralizing antibodies could potentially increase the immunofocusing even more.

In this study, we aimed to identify linear peptide antigens recognized by PfCyRPA-specific neutralizing mAbs to be used as antigens in an effort to focus the immune response. Additionally, we employed the Tag/Catcher-AP205 cVLP platform to facilitate the multivalent presentation of various PfCyRPA-derived protein/peptides, aiming to enhance the overall immunogenicity of the antigen displayed. Our findings underline the effectiveness of a cVLP-based approach in boosting antibody titers and highlight the immunofocusing potential of vaccines consisting of minimal binding regions of a neutralizing monoclonal antibody.

## 2. Materials and Methods

### 2.1. Cloning

PCR with Phusion HF polymerase (Thermo Fisher Scientific, Waltham, MA, USA) was used to subclone SpyCatcher-antigen constructs into the pETmodC for *E. coli* expression and the pTT3 vector for Expi293 expression with either an HIS-tag or C-tag. The constructs were based on a PfCyRPA-SpyCatcher construct (Eurofins Genomics, Ebersberg, Germany) where *N*-glycosylation sites N-X-T/S had been mutated into N-X-A. 

### 2.2. Protein Expression and Purification

For Expi293 expression (Thermo Fisher Scientific), proteins were expressed according to manufacturer’s instructions. Briefly, Expi293 cells were kept at a cell density between 0.3 × 10^6^ and 3 × 10^6^ cells per ml in Expi239 Expression Medium (Thermo Fisher Scientific). Cells were transfected at a density of 3 × 10^6^ cells per ml with 1 µg of plasmid per ml culture diluted in OptiMEM (Thermo Fisher Scientific) and 3.2 µL of Expifectamine (Thermo Fisher Scientific) per ml culture diluted in OptiMEM. On the day following transfection, 6 µL of Enhancer 1 and 60 µL of Enhancer 2 per ml cell culture (Thermo Fisher Scientific) were added. Supernatant containing soluble protein was harvested on day 5 post-transfection and filtered through a 0.22 µm filter.

For *E. coli* expression, transformed BL21 cells were used. A culture was brought to an OD of 0.5–08 at 37 °C, whereafter IPTG was added, and the culture incubated at 20 °C O/N. The culture was harvested, and the bacterial pellet was resuspended and lysed using ultrasound. Soluble protein was isolated using centrifugation and filtered through a 0.22 µm filter.

For C-tag purification, supernatant was loaded on a 1 mL CaptureSelect™ C-tagXL column (Thermo Fisher Scientific), washed with PBS, and eluted with 2 M MgCl_2_ 20 mM TRIS pH 7.4 buffer.

For HIS-tag purification, supernatant was loaded on a 5 mL HisTrap HP column (Cytivalifesciences, Marlborough, MA, USA), washed with binding buffer (25 mM of imidazole, 1 M of NaCl, 20 mM of PO_4_^3−^, pH 8), and gradient-eluted with elution buffer (500 mM of imidazole, 1 M of NaCl, 20 mM of PO_4_^3−^, pH 8).

The purified protein was subsequently concentrated and buffer-exchanged into PBS using 15 mL Amicon centrifugal filters (10 or 30 kDa cutoff) (Sigma-Aldrich, St Lois, MO, USA).

### 2.3. Phage Display

A commercial random peptide library (Ph.D.™-12 Phage Display Peptide Library, NEB, Ipswich, MA, USA) and a gene fragment library generated in our lab were both used for panning against the growth-inhibitory mAb CyP1.9. The gene fragment library was produced and used in panning experiments as described elsewhere [17]. Briefly, DNA encoding PfCyRPA was amplified and digested with DNAseI. Fragments with a size between 50 bp and 250 bp were gel-extracted, fused to the pHEN6 phagemid vector, and transformed into TG1 *E. coli* cells. M13KO7 helper phage was added to express full phages. 

Inhibitory mAb CyP1.9 was bound to magnetic protein G beads (Dynabeads™ Protein G, Invitrogen, Carlsbad, CA, USA) and used for pulldown of phages displaying peptides that were recognized by the mAb. The pulled-down phages were used for re-amplification, and the panning process was repeated twice. Phages yielded by the last round of panning were tested on a single clone basis to confirm their reactivity to CyP1.9 mAb. Positive clones were sequenced, and the insert was aligned with PfCyRPA sequence using Benchling (Benchling.com, San Fransisco, CA, USA). 

For the random peptide library, three rounds of panning with CyP1.9 mAb were performed with the method described above. After isolation and sequencing of positive single phage clones, the newly identified mimotopes were obtained as synthetic peptides and used in a peptide ELISA to validate their binding to CyP1.9 mAb as compared to an isotype control mAb. This extra validation step was required due to the high number of false-positive phage clones observed when using the random peptide library.

### 2.4. Capsid Virus-like Particle Production

Capsid virus-like particles (cVLPs) with SpyTag were essentially produced as described elsewhere [36]. Briefly, the cVLPs were expressed in BL21 *E. coli* as described in “Protein expression and purification”. The supernatant was loaded onto an OptiPrep density gradient (23%, 29%, and 35%) (Sigma-Aldrich), and the cVLPs were purified using ultracentrifugation and subsequently buffer-exchanged into PBS via dialysis using a 1000 kDa cutoff. 

To produce antigen-displaying cVLPs, SpyTag-cVLP was coupled with SpyCatcher-antigen in a 1:2 molar ratio in PBS O/N at 4 °C. *E. coli* endotoxins were removed using Triton X-114 essentially, as described elsewhere [44]. Briefly, 1 mL of coupled cVLPs was mixed with 10 µL Triton X-114, vortexed, and incubated for 5 min on ice, followed by 5 min on heating block set to 37 °C. Next, the detergent phase was separated through centrifugation at 16,000× *g* for 1 min at 37 °C, and the non-detergent supernatant phase was removed. This purification was repeated once. The coupled cVLPs were subsequently purified using ultracentrifugation and buffer-exchanged into PBS as described above.

Coupling efficacy of cVLP:antigen was determined by quantitative densitometry SDS-PAGE. Size distribution of the cVLPs was determined by Dynamic Light Scattering (DLS) analysis (DynaPro Nanostar, Wyatt Technology, Santa Barbara, CA, USA) with measurements taken from 20 acquisitions of 5 s each at 25 °C.

### 2.5. Mice Immunizations

Vaccines diluted in PBS were formulated with AddaVax (InvivoGen, San Diego, CA, USA) for a final concentration of Addavax of 50%, and mice (*n* = 8 per group, all female BALB/c mice) were vaccinated intramuscularly with 50 µL of formulation in each hind leg. For soluble PfCyRPA, 5 µg/125 pmol of protein was used (molar mass of 40 kDa). For PfCyRPA cVLP, 5 µg/0.4 pmol particles were used (molar mass of 12,510 kDa). For other cVLPs, an equal molar amount to the PfCyRPA cVLP was used (0.4 pmol). This resulted in 3.2 µg for Fragment A cVLP, 2.3 µg for MF and Mimotope 1 cVLP, and 2.1 µg for SpyCatcher cVLP. The Fragment A cVLP contained 1.3 µg of unbound Fragment A antigen per 3.2 µg of Fragment A cVLP that was not fused to a cVLP due to incomplete purification.

Mice were immunized on day 0 and boosted on day 28. Sample bleeds were taken on day 27 and full bleeds on day 42. Serum was isolated from clotted blood using centrifugation.

### 2.6. Antibody Purification

Equal volume of serum from each mouse in the immunization group was pooled, mixed with 50 mL of PBS, and loaded onto a 5 mL HiTrap Protein G column (Cytivalifesciences). The column was washed with PBS, IgG-eluted using 0.1 M of glycine pH 2.7, and immediately neutralized with 1 M of TRIS pH 9.

### 2.7. General ELISA Protocol

For enzyme-linked immune-sorbent assay (ELISA), Nunc Maxisorp 96-well plates (Invitrogen) were coated O/N at 4 °C with 2 µg/mL of protein in PBS. TBS 0.05% tween was used as wash buffer. Plates were blocked with blocking buffer (Blocker Casein, Thermo Fisher Scientific, or TBS 0.05% tween 5% skimmed milk) and incubated with a primary antibody followed by a secondary antibody conjugated to either Horseradish peroxidase (HRP) or alkaline phosphatase (AP), diluted in blocking buffer. 

For HRP, plates were developed using 50 µL per well of TMB PLUS2 (Kementec, Taastrup, Denmark) stopped with 50 µL of H_2_SO_4_ and read at 450 and 560 nm using a BioSan HiPo MPP-96 microplate reader. For AP, plates were developed using 100 µL per well of 4-Nitrophenyl phosphate disodium salt hexahydrate tablets (Sigma-Aldrich) dissolved in 1× Diethanolamine Substrate Buffer (Sigma-Aldrich) and read at 405 and 560 nm. For both 405 nm and 450 nm signals, the corresponding 560 nm signal was subtracted.

### 2.8. Serum ELISA

Plates were coated with PfCyRPA and developed with AP, as described above. Serum from each individual mice was used as primary antibody in 12 three-fold dilutions starting at 1 in 50, along with a positive control of anti-PfCyRPA mAb CyP1.9 held at 0.035 µg/mL. An anti-mouse IgG (γ-chain specific, Sigma-Aldrich) antibody coupled to alkaline phosphatase was used as secondary antibody. The plate was read when OD405 of the positive control reached OD 1. The OD405 of each well was subtracted with the OD560 of each well, and the mean value of three blank wells was subtracted from all wells of the same plate. Area under the curve (AUC) was calculated based on the OD405 minus background plotted against log10 to the inverse of the serum dilution.

### 2.9. Quantification of Serum IgG Concentration

Plates were coated with an anti-mouse IgG (γ-chain specific, Sigma-Aldrich) antibody. Serum was titrated in 8 two-fold dilutions starting at 1 in 10,000. Purified polyclonal mouse IgG was titrated in 8 two-fold dilutions starting at 100 ng/mL. Polyclonal anti-mouse IgG conjugated to HRP was used as detection antibody. A standard curve was made with linear regression for each plate based on the linear part of the reaction (OD450 < 1). Serum IgG per mouse was calculated based on the mean of all dilutions of the serum titration that had OD450 within the OD450 used for the standard curve. 

### 2.10. Quantification of PfCyRPA Specific IgG by ELISA

The ELISA was prepared and developed as described in the section “Serum ELISA”. Instead of serum, we used pooled, purified IgG for each group. Linear regression was used to determine the IgG concentration that yielded OD 1. GIA data were normalized to this value by dividing the IgG concentration used in the assay by the IgG concentration that yielded OD 1; e.g., a GIA measured at 1000 µg/mL for a group that reached OD 1 at 0.39 µg/mL would be normalized to 1000/0.39 = 2564 AU.

### 2.11. Polyclonal IgG-Binding Kinetics Determined Using Attana © Biosensor

Kinetic interaction experiments of PfCyRPA antigens binding to polyclonal total IgG (pAbs) from vaccinated mice were performed using a quartz crystal microbalance (QCM) biosensor Attana Cell A200 instrument (Attana, Stockholm, Sweden). Recombinant PfCyRPA was diluted to 100 μg/mL in 10 mM of Na-acetate pH 4.5 and immobilized on a chip (sensor chips low non-specific binding (LNB)-Carboxyl 3623-3103, Attana, Stockholm, Sweden) via amine coupling using EDC and S-NHS chemistry, following manufacturer’s instructions (Amine Coupling kit 3501-3001, Attana). The chip was pre-wetted with ultra-pure water before the chip surface was activated with EDC/sulfo-NHS 2. The antigen was then immobilized on the activated chip surface, and afterward, ethanolamine was used to deactivate the surface. A non-coated LNB chip was used as reference. Seven serial 2-fold dilutions of pAbs were prepared in HBS-T buffer starting at the concentration of 100 µg/mL (660 nM). The PfCyRPA-immobilized chip was regenerated using 10 mM of glycine-HCl pH 1.5.

All sensorgrams were recorded at 25 μL/min at 22 °C using an 84 s association and 300 s dissociation time and then regenerated to allow complete baseline recovery. Injection of running buffer was subtracted for each sensorgram, and the kinetic parameters k_on_ and k_off_ were fitted with a bivalent interaction model using TraceDrawer 1.9.2 (Ridgeview Instruments, Uppsala, Sweden).

### 2.12. Growth Inhibition Activity Assay

The efficacy of the IgG of immunized mice to prevent parasite invasion of red blood cells was determined using the standardized growth inhibition activity (GIA) assay as described elsewhere [16]. Briefly, 3D7 *P. falciparum* was kept in culture using fresh O+ erythrocytes at 2% hematocrit. Mid-trophozoite stage parasites were mixed with purified IgG; positive control, mAb R5.016; a negative control, mAb; media; or 5 mM of EDTA. Parasites were incubated for 48 h corresponding to one life-cycle, and the amount of infected red blood cells was determined by a lactate dehydrogenase assay, where the signal was read at 630 nm. Percentage growth inhibition was calculated as
$$\%\mathrm{GIA}=100-\frac{A630\;\mathrm{inhibited}\;\mathrm{sample}-A630\;\mathrm{RBCs}\;\mathrm{only}}{A630\;\mathrm{negative}\;\mathrm{control}-A630\;\mathrm{RBCs}\;\mathrm{only}}\times 100$$

GIA was performed for 6 dilutions in triplicates in minimum of three independent experiments, with exceptions where insufficient amounts of IgG were available. These exceptions are priming with soluble PfCyRPA and boosting with either Fragment A cVLP or Mimotope 1 cVLP run at 1000 µg/mL, where only two independent experiments were made, and priming with soluble PfCyRPA and boosting with SpyCatcher cVLP, where all concentrations were only run in two independent experiments. 

Outliers were excluded based on the modified Z-score, as explained elsewhere [45].

### 2.13. Data Analysis and Visualization

Data analysis was performed in R version 4.2.3 using RStudio version 2023.03.0 + 386 and GraphPad Prism version 10.2.2. For all comparisons between soluble PfCyRPA and PfCyRPA displayed on a cVLP, a two-tailed Mann–Whitney test was used. For comparisons between the cVLP displayed vaccines, a Kruskall–Wallis test with Dunn’s post test was used.

## 3. Results

### 3.1. The Neutralizing mAb CyP1.9 Recognizes a Peptide Fragment of PfCyRPA and a 12-mer Mimotope

Neutralizing monoclonal antibodies (mAbs) targeting PfCyRPA are known to bind the blade two and three regions of the protein, known as Fragment A. In order to focus the immune response toward these specific highly neutralizing epitopes, we set out to identify a short linear peptide that can be recognized by the neutralizing mAb CyP1.9. Using such peptides for immunizations could potentially greatly focus the immune response toward a neutralizing epitope. To that end, we generated a phage display library consisting of fragments of PfCyRPA encoding DNA of about 50–300 bp, resulting in protein fragments of approx. 15 to 100 amino acids. 

Panning the phage library with CyP1.9 on the library resulted in an increased reactivity of the mAb against the phage pool measured by an enzyme-linked immunosorbent assay (ELISA), especially after the third round of panning (Figure 1a). By testing the reactivity toward single phage clones, also by ELISA, several phage clones were identified that expressed peptides only recognized by CyP1.9 and not by an isotype control antibody (Figure 1b). Multiple phages were sequenced and aligned to the PfCyRPA DNA sequence, where all were aligned between blades two and three of PfCyRPA (Figure 1c). A minimal fragment (MF) was defined from the smallest possible overlap between the peptides (Figure 1c,f).

An alternative approach to identify small peptides resembling epitopes is panning on random peptide phage display libraries. We here panned the CyP1.9 mAb on a 12-mer random peptide library, which yielded increased reactivity toward the phage pool between rounds 1 and 3 of panning (Figure 1d). Three peptides recognized by CyP1.9 were identified. However, when validating the binding capacity of the three potential epitope mimics (mimotopes) as biotinylated synthetic peptides by ELISA, only one (Mimotope 1) was recognized by CyP1.9 (Figure 1e). Aligning MF with Mimotope 1 showed an overlap between the two amino acid sequences (Figure 1f). 

### 3.2. PfCyRPA-Derived Antigens Were Conjugated to cVLPs and Were Recognized by Neutralizing mAbs

Having a selection of antigens ranging from full-length PfCyRPA to Fragment A, MF, and Mimotope 1, the next step was to conjugate the antigens to capsid virus-like particles (cVLPs) to enhance their immunogenicity. All vaccine designs are depicted in Figure 2a. cVLP antigen display was facilitated by mixing the different antigens (genetically fused to SpyCatcher) with AP205-cVLP displaying one SpyTag per cVLP subunit. A SpyTag-AP205-cVLP coupled with SpyCatcher alone was produced to serve as a negative control.

cVLP-based vaccine formulations were quality-checked by dynamic light scattering (DLS) analysis to assess for potential aggregation. DLS analysis of cVLPs displaying PfCyRPA, MF, Mimotope 1, and SpyCatcher showed a single peak representing monodisperse particles with a radius of ~25 nm (Figure 2b). DLS analysis of cVLP displaying fragment A showed a single peak with a radius of ~100 nm, suggesting some agglomeration of antigen:cVLP complexes (Figure 2b). 

The antigen coupling efficacy, defined as the percentage of cVLP subunits that were coupled via Tag/Catcher conjugation to the antigen, was determined by densitometric SDS-PAGE analysis. These results suggested that the coupling efficiency was 100% for all cVLPs, as no unbound cVLP subunit protein could be detected (Appendix A). However, for Fragment A, uncoupled protein Fragment A was isolated alongside the fully coupled cVLPs corresponding to 29% of the total protein mass, indicating a non-covalent association of antigen to the Fragment A cVLPs. Soluble PfCyRPA purity was determined by SDS-PAGE, where no additional bands than the one corresponding to full-length protein were observed (Appendix A).

A panel of neutralizing and non-neutralizing PfCyRPA-specific mAbs were tested for reactivity toward soluble PfCyRPA and the cVLP vaccines using ELISA (Figure 2c). The reactivity for all non-neutralizing mAbs was abolished on the immunofocusing constructs Fragment A, MF, and Mimotope 1, indicating the removal of non-neutralizing epitopes. Neutralizing mAbs CyP2.27 and CyP2.39 had no reactivity toward MF and Mimotope 1 but maintained a decent reactivity toward the larger Fragment A. Neutralizing mAb CyP1.9 maintained reactivity toward all constructs except the negative control SpyCatcher cVLP. However, the reactivities for Fragment A and Mimotope 1 were diminished compared to soluble PfCyRPA. The reactivity of CyP1.9 toward MF was greatly reduced, indicating that important residues had been removed.

### 3.3. Displaying PfCyRPA on cVLP Increases Antigen-Specific IgG Production

To compare the immunogenicity of the different vaccine constructs, eight mice per group were immunized with the purified cVLPs in a homologous prime/boost regimen (Figure 2d and Appendix A). A group vaccinated with soluble PfCyRPA was included for comparison, and a group vaccinated with naked SpyCatcher cVLP was included as a negative control. 

Serum IgG reactivity toward PfCyRPA antigen of all individual vaccinated mice was determined by ELISA using serum obtained on day 27 (one day prior to boost) and day 42 (14 days post-boost) (Figure 2e). Serum samples from the PfCyRPA cVLP-vaccinated group displayed a significantly higher anti-PfCyRPA IgG titer compared to soluble PfCyRPA-vaccinated mice (*n* = 8). While fragment A cVLP induced a minor response, sera from MF cVLP- and Mimotope 1 cVLP-vaccinated animals displayed low to no reactivity, similar to what was observed for the negative control group vaccinated with SpyCatcher cVLP (Figure 2e). The serum response of mice prebleeds prior to vaccinations is shown in Appendix A.

Next, we measured the serum IgG concentration using ELISA (Figure 2f). The mean serum IgG concentration of the vaccine groups was 1.3 ± 0.3 mg/mL for soluble PfCyRPA, 2.4 ± 0.4 mg/mL for PfCyRPA cVLP, 1.1 ± 0.4 mg/mL for Fragment A cVLP, 1.6 ± 0.3 mg/mL for MF cVLP, 1.8 ± 0.4 mg/mL for Mimotope 1 cVLP, and 1.2 ± 0.2 mg/mL for SpyCatcher cVLP (± signifies standard deviation). Thus, the group vaccinated with PfCyRPA cVLP had 1.8 times more IgG than the group vaccinated with soluble PfCyRPA. The serum IgG concentration of mice prebleeds prior to vaccinations is shown in Appendix A.

The serum samples were pooled for each group, and the serum IgG was purified. The amount of antigen-specific IgG per total IgG was measured by ELISA and calculated as the amount of total IgG needed for OD1 using linear regression (Figure 2g and Appendix A). The higher the fraction of PfCyRPA-specific IgG, the lower the total IgG needed in the assay for OD1. Soluble PfCyRPA and PfCyRPA cVLP showed similar amounts of antigen-specific IgG, followed by Fragment A and MF cVLP, indicating that the truncation of PfCyRPA reduced the amount of PfCyRPA-specific IgG. Mimotope 1 cVLP showed a similar response to the negative control, SpyCatcher cVLP. For groups MF cVLP, Mimotope 1 cVLP, and SpyCatcher cVLP, insufficient IgG was available to reach OD1. OD1 was then inferred from a linear regression based on the available data.

The purified IgG was tested for neutralizing capabilities using the growth inhibition activity (GIA) assay. PfCyRPA cVLP and soluble PfCyRPA display similar GIA when measured by the total IgG used in the assay, both reaching 92% inhibition at a 1000 µg/mL total IgG concentration (Figure 2h, left panel). Fragment A cVLP and MF cVLP reached 34% and 44% inhibition, respectively, at the same µg/mL total IgG concentration. Mimotope 1 performed similarly to the negative control, SpyCatcher cVLP (Figure 2h, left panel).

To mimic serum dilutions in the performed GIA, the total IgG used in the assay was divided by the mean total serum IgG concentration of the corresponding group (Figure 2e); e.g., 100 µg/mL total IgG in a group with a mean serum IgG concentration of 1000 µg/mL would yield a normalized value of 10^−1^, corresponding to a 1 in 10 serum dilution. In this way, the neutralizing capabilities of IgG were kept free from other variables, such as the complement system, while still accounting for differences in the IgG concentration in the serum. When normalizing to the serum IgG concentration, PfCyRPA cVLP outperforms soluble PfCyRPA (Figure 2h, middle panel). The remaining groups keep their relative position to soluble PfCyRPA, as depicted in Figure 2h, left panel.

In the GIA assay, it was evident that using full-length PfCyRPA, either as a soluble protein or displayed on a cVLP, was superior to the truncated constructs Fragment A cVLP, MF cVLP, and Mimotope 1 cVLP. This indicated that we had not achieved any absolute immunofocusing, i.e., higher levels of neutralizing IgG per total IgG or per serum dilution. To examine whether we had achieved relative immunofocusing, i.e., higher levels of neutralizing IgG among the antigen-specific IgG fraction, we normalized the total IgG used in the GIA assay to the amount of antigen-specific IgG in the IgG people, as seen in Figure 2g, thereby depicting GIA per arbitrary PfCyRPA specific IgG unit (Figure 2f, right panel). 

When plotting GIA per PfCyRPA-specific IgG, MF cVLP appears to clearly outperform all other constructs at lower antibody levels, which could signify the induction of PfCyRPA-specific antibodies with superior neutralizing capacity. However, far lower PfCyRPA titers were reached. PfCyRPA cVLP and soluble PfCyRPA performed similarly to each other, while Fragment A showed a similar amount of GIA per arbitrary unit PfCyRPA specific IgG without reaching the same neutralizing effect as PfCyRPA cVLP (Figure 2f, right panel).

### 3.4. Boosting with Linear Epitopes Selectively Increases the Induction of Neutralizing Antibodies

With the aim to increase the neutralizing potential by focusing the vaccine-induced antibody response, we tested a heterologous prime/boost immunization regimen (*n* = 8 per group, except Mimotope 1 cVLP, where *n* = 7). This regimen involved administering soluble PfCyRPA as the prime vaccination followed by a booster vaccination using the cVLP-displayed antigen vaccines (see Figure 3a and Appendix A). The characterization of the vaccine-induced antibody responses was performed as previously described for the homologous vaccine regimen. 

Boosting with PfCyRPA cVLP provided the highest PfCyRPA-specific reactivity among the cVLP boosts and the highest total IgG production, albeit in both cases lower than the PfCyRPA cVLP homologous prime/boost (Figure 3b,c). The immunofocusing PfCyRPA constructs yielded low amounts of serum IgG concentration compared to the homologous prime/boost. However, compared to boosting with the negative control, SpyCatcher cVLP provides a small, albeit not statistically significant, increase in antigen-specific titers.

When depicting GIA per total IgG used in the assay (Figure 3e, left panel), boosting with PfCyRPA cVLP achieves the highest degree of inhibition at 83%, though not as high as the homologous prime/boost with PfCyRPA cVLP. The inhibition achieved by boosting with Fragment A cVLP and MF cVLP is 50% and 41%, respectively, at a 1000 µg/mL total IgG concentration, performing similarly to the homologous prime/boost setting for these cVLPs vaccines. Mimotope 1 cVLP is greatly improved by the soluble PfCyRPA prime and reaches its maximum inhibition of 43% at a 500 µg/mL total IgG concentration. In comparison, priming with soluble PfCyRPA and boosting with SpyCatcher cVLP achieves 18% inhibition at 1000 µg/mL IgG. Thus, the increase in GIA is likely due to antibody responses directed by the immunofocusing constructs.

Depicting GIA per total IgG normalized to serum IgG concentration instead of the total IgG (Figure 3e, middle panel) created the greatest change in the heterologous PfCyRPA cVLP due to its higher IgG production. It is, however, still not on par with homologous prime/boost with PfCyRPA cVLP.

Finally, plotting GIA per arbitrary unit PfCyRPA-specific IgG (Figure 3e, right panel) shows a higher degree of inhibition when boosting with Fragment A, MF, and Mimotope 1 cVLPs at lower amounts of PfCyRPA-specific IgG compared to PfCyRPA cVLP, indicating a relative, though not absolute, immunofocusing. However, homologous prime/boost with MF cVLP yields the highest degree of neutralization per PfCyRPA-specific antibody, as seen by the far lower EC30.

### 3.5. Immunization with PfCyRPA cVLP Induces the Lowest EC30 When Normalized to Serum IgG Concentration

To visualize the differences in the parasite growth inhibition between all 11 vaccination groups, EC30 values were calculated for all applicable groups from the GIA measured by total IgG (Figure 3f left panel), total IgG normalized to the serum IgG concentration (Figure 3f middle panel), and PfCyPRA-specific IgG (Figure 3f right panel). Measured using the total IgG, homologous prime/boost with soluble PfCyRPA and PfCyRPA cVLP performed similarly with an EC30 of 53.2 ± 16.7 µg/mL and 24.7 ± 18.1 µg/mL, respectively (± signifies the standard error of the mean). However, when normalizing to the amount of IgG induced by each vaccine, soluble PfCyRPA has an EC30 of 0.041 ± 0.013 AU and PfCyRPA cVLP an EC30 of 0.010 ± 0.007 AU, as the latter induced 1.8 times higher amounts of IgG. When looking at PfCyRPA-specific IgG, homologous prime/boost with MF cVLP appears to drastically outperform all other candidates, with an EC30 of 0.12 ± 0.08 AU compared to 152.58 ± 111.98 AU for soluble PfCyRPA.

### 3.6. Immunization with PfCyRPA cVLP Induces Higher IgG2a, IgG2b, and IgG3 Responses Compared to Soluble PfCyRPA

As soluble PfCyRPA and PfCyRPA cVLP overall induced the most PfCyRPA-specific antibodies, we next examined the differences in Ig isotypes and IgG subclasses specific to PfCyRPA in serum using ELISA (Figure 4a). The two vaccines induced similar amounts of IgG1, but PfCyRPA cVLP induced significantly more IgG2a, IgG2b, and IG3 than soluble PfCyRPA (*p*-value < 0.001 for all subclasses except IgG1). No detectable IgM targeting PfCyRPA was present in samples from either group. 

To further characterize the antibody response induced by PfCyRPA and PfCyRPA cVLP, we performed a kinetic analysis of the interaction between the vaccine-induced polyclonal IgG and PfCyRPA antigen (Figure 4b). The antigen was immobilized on an LNB carboxyl chip, and the two IgG pools were tested in a concentration titration series using an Attana Biosensor. Overall, the IgG samples showed similar kinetic parameters in their binding to the antigen, with IgG from the PfCyRPA cVLP group displaying a slightly slower dissociation rate (kd1values: 1.17 × 10^−3^ for IgG from the soluble PfCyRPA group; 9.53 × 10^−4^ for IgG from the PfCyRPA cVLP group). 

## 4. Discussion

PfCyRPA is a highly conserved [1,9,10,11] subunit of the pentameric complex PCRCR [1,2,3,4,5,6,7] and has been shown to be essential for parasite survival and RBC invasion [1,6,8]. Further, antibodies targeting PfCyRPA have been shown to inhibit this invasion [1,4,12,13,14,15,16]. For these reasons, PfCyRPA is regarded as a promising vaccine antigen. Another component of the PCRCR complex, PfRH5, has been tested in clinical trials but failed to provide protection, likely due to insufficient titers of neutralizing antibodies targeting the overall complex. To enable the PCRCR complex as an antigen for an effective malaria vaccine, we need to develop strategies to increase its immunogenicity and antigenicity to achieve the maximum induction of neutralizing antibodies. This study explores two strategies to accomplish this for the PCRCR component PfCyRPA: immunofocusing via antigen design and cVLP display.

Immunization with cVLP-displayed PfCyRPA resulted in significantly higher (1.8 fold) anti-PfCyRPA titers compared to immunization with soluble PfCyRPA. Further, cVLP antigen display resulted in the IgG subclass switching from primarily murine IgG1 to a balanced IgG1, IgG2a, and IgG2b response, as well as an IgG3 response. This switching is likely an effect of single-stranded bacterial RNA that is encapsulated during the production of the AP205 cVLP and stimulates TLR7/8 receptors [46,47,48]. Overall, these results align with multiple previous studies showing that the Tag/Catcher-Ap205 platform can increase the immunogenicity of displayed protein antigens and alter the IgG subclass profile [36,37,39,49]. 

The neutralizing effect measured by the total IgG concentration was almost identical between the groups immunized with soluble PfCyRPA and PfCyRPA cVLP. This indicates that the fraction of the immune response targeting the cVLP platform had little relevance. However, the similar neutralizing effect of equal total IgG concentrations did not reflect the difference in PfCyRPA-specific serum titers. Upon investigation, the difference was found to be due to a far higher induction of serum IgG when utilizing the cVLP platform over soluble protein. When measuring the neutralizing effect created by IgG normalized to the serum IgG concentration, the cVLP outperformed soluble protein. This is an important point when comparing vaccine-induced Abs efficacy since using purified IgG for neutralization assays instead of serum removes unwanted variables but does not factor in differences in serum IgG concentrations, which can significantly bias the results. It especially warrants caution when testing different delivery systems in vivo, such as mRNA or protein-based vaccines, changes in adjuvant or displaying methods such as cVLPs, as well as large differences in vaccine dose, as these differences potentially could result in very different amounts of IgG production.

The immunofocusing constructs Fragment A, MF, and Mimotope 1 maintained a recognition by neutralizing mAbs while abolishing recognition by non-neutralizing mAbs. Mimotope 1 was surprisingly better recognized by CyP1.9 than MF, indicating that MF lacked important residues. Fragment A lost some recognition by neutralizing mAbs CyP2.27, CyP2.38, and CyP2.39, either due to the removal of important residues upon truncation or distortion of the protein fold. The latter is an important point indicating that a protein with distinctly separated domains is more easily truncated than a compact one such as PfCyRPA.

Homologous prime/boost with the immunofocusing constructs showed a strong neutralizing effect measured by GIA in the group vaccinated with MF cVLP when considering that the PfCyRPA-specific response was only measurable when using high concentrations of IgG. This indicates that a small but very focused and efficient antibody response had been induced, as seen by the low EC30 when measuring by PfCyRPA-specific antibodies. In comparison, Fragment A had a higher PfCyRPA-specific response but yielded the same inhibition by PfCyRPA-specific antibodies as full-length soluble PfCyRPA. 

Priming with soluble PfCyRPA and boosting with the immunofocusing constructs led to only small increases in PfCyRPA-specific serum titers compared to boosting with the negative control SpyCatcher cVLP. However, it resulted in clear differences in GIA, indicating that the increased PfCyRPA response had a highly neutralizing effect. PfCyRPA cVLP outperforms the immunofocusing constructs when measuring by the total IgG concentration and when normalizing to the serum IgG concentration, signifying that albeit a relative immunofocusing had been achieved, the absolute amount of neutralizing antibodies had not been increased.

Similar results have been observed for PfRH5, where a neutralizing epitope was grafted onto a protein scaffold. This produced higher GIA per antigen-specific IgG but not per total IgG [50]. It was observed that vaccination with full-length PfRH5 and the epitope scaffold protein yielded similar epitope-specific responses, indicating that there is a limit to the epitope-specific responses that can be achieved. This gives rise to the question of whether immunofocusing is a viable way of achieving higher absolute titers of neutralizing antibodies. 

Truncation of protein has, however, been shown to be able to provide an absolute immunofocusing effect. When immunizing with PfRH5, PfCyRPA, and PfRipr, PfRipr dominated the immune response and ultimately led to an inferior neutralizing effect than when immunizing with PfRH5 alone [51]. By removing a large part of PfRipr, shown to primarily harbor non-neutralizing epitopes, the immune response against PfRH5 and PfCyRPA was restored and an overall higher neutralizing effect was achieved. Instead of trying to focus the immune response against a small, neutralizing epitope, as the immunofocusing constructs presented here and the PfRH5 epitope scaffold described above, truncation of PfRipr exploited that an apparent limit of the overall antibody response had been reached, and the removal of a large surface containing non-neutralizing epitopes could free up resources for PfRH5 and PfCyRPA. 

The truncated PfRipr consisted of distinct, epidermal growth factor (EGF)-like domains, enabling truncation without distortion of the neutralizing epitopes. Thus, immunofocusing through truncation is heavily dependent on the ability to maintain the fold of the protein, as well as having a scenario where non-neutralizing epitopes dominate the immune response. In the case of PfCyRPA, no increase in neutralizing titers was observed for antigens MF, Mimotope 1, and Fragment A compared to full-length PfCyRPA. This indicates either that the non-neutralizing epitopes contained in PfCyRPA do not constitute a problem for the focus of neutralizing response or that the truncation led to both the removal of non-neutralizing epitopes as well as distortion of neutralizing epitopes. 

## 5. Conclusions

Raising the neutralizing antibody titers against the essential blood-stage invasion complex PCRCR remains the main goal for creating an efficient blood-stage malaria vaccine. In this study, we have shown that presenting the promising malaria vaccine target PfCyRPA antigen on cVLPs leads to an increase in the induction of total serum IgG and parasite invasion-neutralizing antibodies. Further, the immunofocusing we employed here increased the quality of the antigen-specific Ab response but did not result in higher overall neutralizing antibody titers. However, it was shown that truncation of the PCRCR protein PfRipr leads to an enhanced neutralization capacity of the antibody response. The nature and size of the antigen, as well as the epitope landscape with regard to neutralization and immunodominance, should therefore be considered prior to employing immunofocusing strategies. When, as with PfCyRPA and PfRH5, a large part of the protein contains neutralizing epitopes, immunofocusing may in fact remove more neutralizing epitopes than is beneficial.

## Figures and Tables

**Figure 1 vaccines-12-00859-f001:**
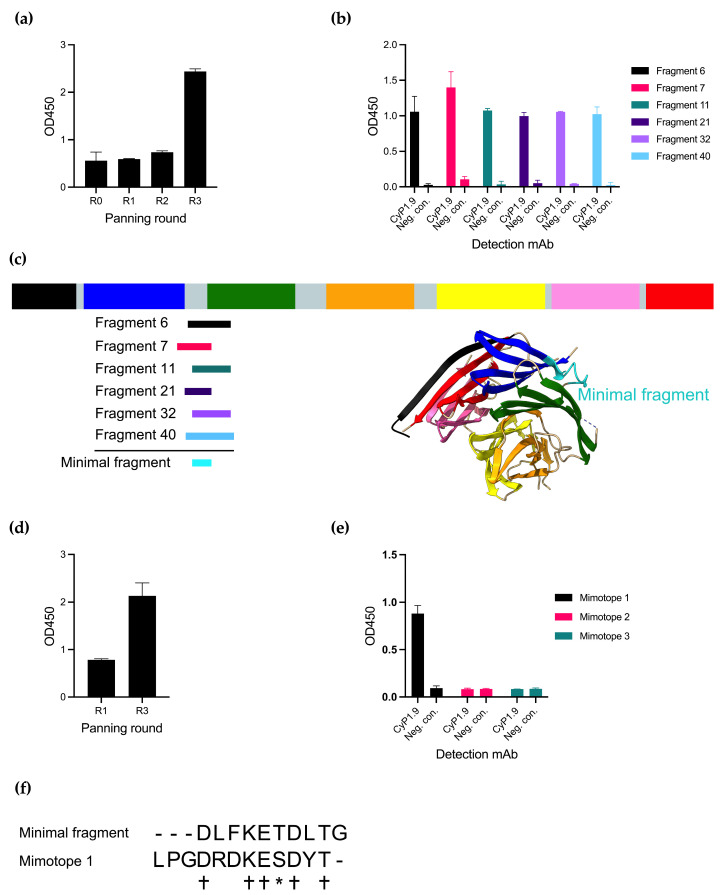
Identifying peptides binding to inhibitory mAb CyP1.9. (**a**) Reactivity of inhibitory mAb CyP1.9 against phage pools derived from a PfCyRPA gene fragment library after panning rounds zero through three measured by ELISA. (**b**) Reactivity of inhibitory mAb CyP1.9 and isotype control against single phage clones derived from the random peptide library after panning round three measured by ELISA. (**c**) Insert of sequenced single clone phages derived from the gene fragment library aligned to PfCyRPA. PfCyRPA, with its six blades, is shown as a primary structure sketch and tertiary protein structure (PDB 5TIK), with the minimal binding fragment (MF) shown in cyan. (**d**) Reactivity of inhibitory mAb CyP1.9 phage pools derived from the Ph.D.-12 phage display peptide library (NEB) after panning rounds one and three measured by ELISA. (**e**) Reactivity of inhibitory mAb CyP1.9 and isotype control against synthesized peptides derived from the Ph.D.-12 phage display peptide library isolated after panning round three measured by ELISA. (**a**,**b**,**d**,**e**): For all ELISA-based panels, mean and SD of OD450 measured in duplicates are shown. (**f**) MF aligned to Mimotope 1. (+) denotes identical amino acid, (*) denotes amino acid with similar biochemical properties, (-) denotes no amino acid.

**Figure 2 vaccines-12-00859-f002:**
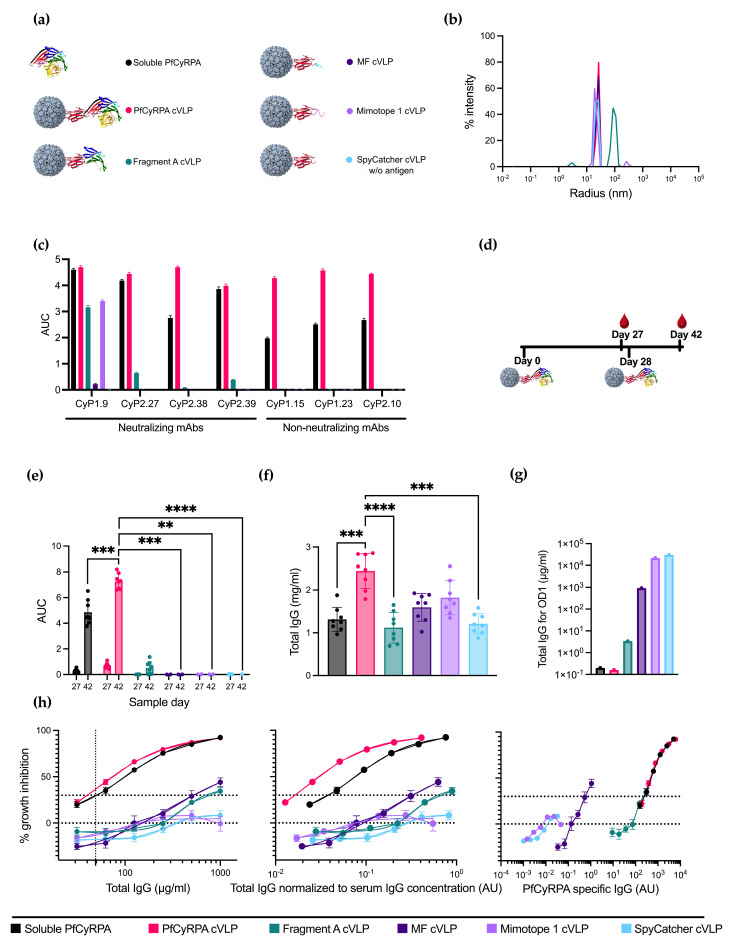
Design, production, and immunizations with PfCyRPA-based vaccines. (**a**) Six different protein constructs were used for immunizations. One was soluble PfCyRPA protein, and the remaining were five different antigens coupled to a cVLP using the SpyTag/Catcher system. These antigens included full-length PfCyRPA, Fragment A of PfCyRPA, MF, Mimotope 1, and a naked SpyCatcher-fused cVLP included as a negative control. (**b**) Radius of the cVLPs determined by dynamic light scattering (DLS). On the *x*-axis is the radius in nm, and on the *y*-axis is the intensity. (**c**) Reactivity of neutralizing and non-neutralizing mAbs against the vaccine constructs is measured as AUC. (**d**) Vaccine regimen. Mice were primed on day 0 and boosted on day 28 with the same vaccine. Blood samples were collected on days 27 and 42. (**e**) AUC of serum titrations against PfCyRPA of individual mice (*n* = 8 per group). Sample day is denoted on the *x*-axis. Mean and SD of each group are shown. (**f**) Serum IgG concentration in mg/mL per mouse was determined using ELISA. (**g**) Concentration of IgG (µg/mL) purified from pooled serum needed for OD1 measured by ELISA. (**h**) Growth-inhibition activity of purified IgG from pooled serum shown by total IgG (µg/mL), IgG normalized to serum IgG concentration (AU), and PfCyRPA-specific IgG (AU). Growth-inhibition activity was measured at each total IgG concentration in triplicates in at least three different experiments, with exceptions due to lack of IgG shown in the Methods section. Mean and SEM are shown. Values with a modified z-score of >3.5 were excluded as outliers. For panel (**e**,**f**), soluble PfCyRPA was compared to PfCyRPA cVLP using a two-tailed Mann–Whitney test to test for the isolated effect of cVLP display. For panel (**e**), day 42 was tested. The immunofocusing constructs Fragment A cVLP, MF cVLP, and Mimotope 1 cVLP, as well as negative control SpyCatcher cVLP, were compared to PfCyRPA cVLP using a Kruskal–Wallis test with Dunn’s post-test to test for the isolated effect of modifications of the displayed antigen. ** = *p* ≤ 0.01, *** = *p* ≤ 0.001, **** = *p* ≤ 0.0001. Calculated comparisons that were not statistically significant are not shown.

**Figure 3 vaccines-12-00859-f003:**
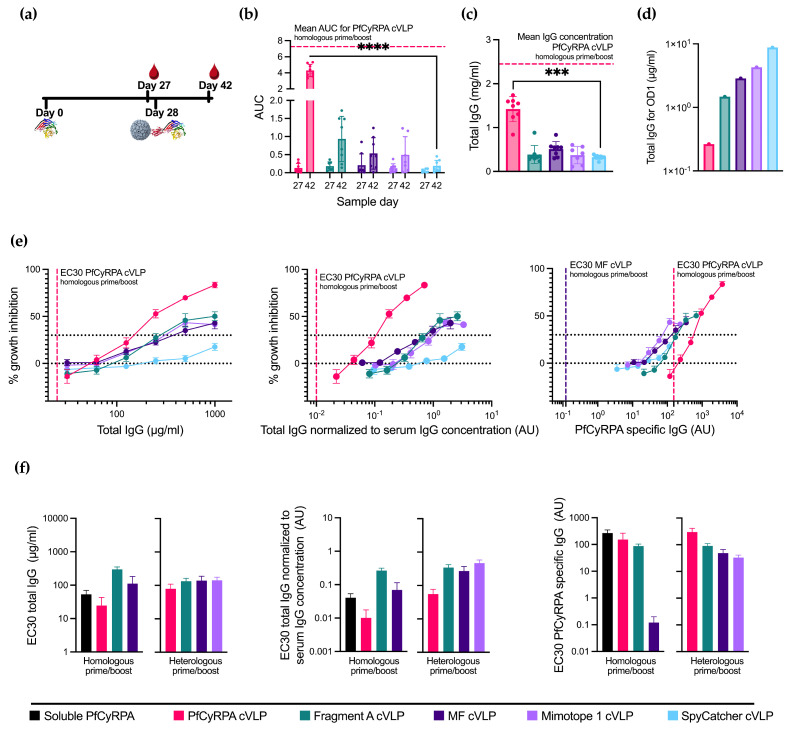
Priming with soluble PfCyRPA alters cVLP immunization efficacy. (**a**) Vaccine regimen. Mice were primed on day 0 with soluble PfCyRPA and boosted on day 28 with one of five cVLPs presented in Figure 2. Blood samples were taken on days 27 and 42. (**b**) AUC of serum titrations against PfCyRPA of individual mice (*n* = 8 per group, except Mimotope 1 cVLP, where *n* = 7). Sample day is denoted on the *x*-axis. Mean and SD of each group shown. Mean of AUC of PfCyRPA cVLP from Figure 2 is shown in dashed line. (**c**) Serum IgG concentration in mg/mL per mouse at day 42 determined by ELISA. Mean of serum IgG concentration of PfCyRPA cVLP from Figure 2 is shown in dashed line. (**d**) Concentration of IgG (µg/mL) purified from pooled serum needed for OD1 measured by ELISA. (**e**) Growth-inhibition activity of purified IgG from pooled serum shown by total IgG (µg/mL), IgG normalized to serum IgG concentration (AU), and PfCyRPA-specific IgG (AU). EC30 of PfCyRPA cVLP is shown in dashed line. EC30 of MF cVLP showed in dashed line when plotted against PfCyRPA-specific antibody. Growth-inhibition activity was measured at each total IgG concentration in triplicates in at least three different experiments, with exceptions due to lack of IgG shown in Results section. Mean and SEM are shown. Values with a modified z-score of >3.5 were excluded as outliers. (**f**) EC30 values are calculated from graphs depicted in Figure 2f and 3D using the function “[Agonist] vs. response–Find ECanything” in Graphpad Prism. EC30 is only reported for groups that reached 30% inhibition. For panel (**b**,**c**), boosting with the immunofocusing constructs was tested against negative control SpyCatcher cVLP using a Kruskal–Wallis test with Dunn’s post-test. For panel (**b**), day 42 was tested. *** = *p* ≤ 0.001, **** = *p* ≤ 0.0001. Calculated comparisons that were not statistically significant are not shown.

**Figure 4 vaccines-12-00859-f004:**
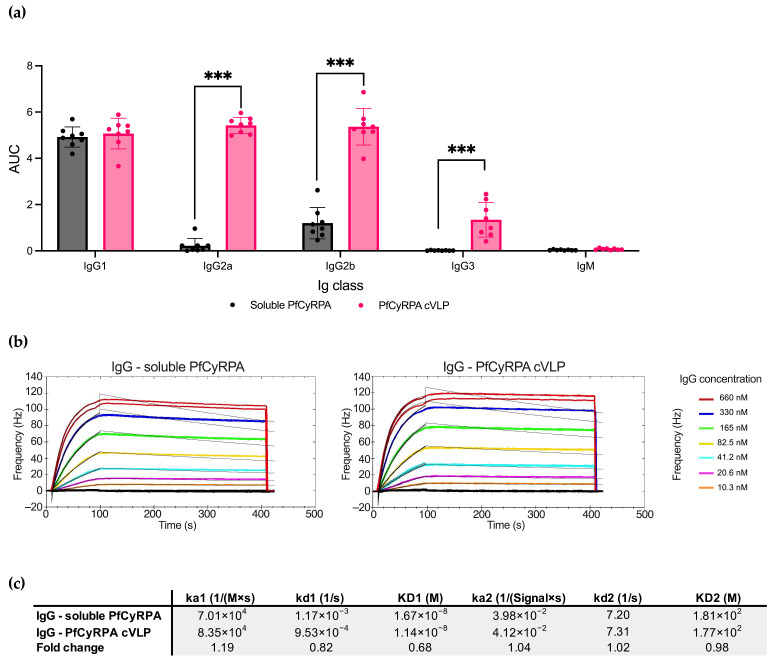
Soluble PfCyRPA and PfCyRPA cVLP exhibit differences in IgG subclass response but similar antigen-binding kinetic parameters. (**a**) AUC of serum titrations against PfCyRPA of individual mice (*n* = 8 per group) measured by Ig class and IgG subclass by class-specific secondary antibody. Mean and SD shown. Statistics were performed using a two-tailed Mann–Whitney test. *** = *p* < 0.001. (**b**) Sensorgrams of purified polyclonal IgG from mice vaccinated with soluble PfCyRPA or PfCyRPA displayed on a cVLP. IgG flowed over PfCYRPA antigen that had been immobilized on a low non-specific binding (LNB)-Carboxyl chip. Fitted nonlinear regression curves based on a bivalent model are depicted as black lines overlaying the experimental data. (**c**) Calculated Ka, Kd, and KD values based on the bivalent model are depicted. ka1 [1/(M × s)]: The first association rate constant; kd1 [1/s]: The first dissociation rate constant; ka2 [1/([SignalUnit] × s)]: The second association rate constant; kd2 [1/s]: The second dissociation rate constant. Fold change depicted.

## Data Availability

All data are available upon request.

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
