# Peer review of "Leveraging Immunofocusing and Virus-like Particle Display to Enhance Antibody Responses to the Malaria Blood-Stage Invasion Complex Antigen PfCyRPA"

_vaccines, 2024, doi:10.3390/vaccines12080859_

Round 1

Reviewer 1 Report

Comments and Suggestions for Authors

The PCRCR complex is regarded as an antigen for an effective malaria vaccine, but there is  need to develop strategies to increase its immunogenicity and antigenicity to achieve the highest induction of neutralizing antibodies. Authors explores two strategies to accomplish this for the PCRCR component PfCyRPA: Immunofocusing via antigen design, and cVLP display. While immunofocusing may have increased the quality of the antigen-specific Ab response, it did not achieve overall higher neutralizing titers. In this study it was shown that presenting a highly relevant PfCyRPA antigen on cVLPs leads to an increase in the induction of total antigen-specific antibody titers. 

The authors underline  the potential of a cVLP based malaria vaccine including full-length PfCyRPA, which could be combined with other leading  malaria vaccine antigens, presented on cVLPs.

I have found presented results as significant and recommend for publication. 

Author Response

We thank reviewer 1 for their time and their peer-review of our manuscript.

Reviewer 2 Report

Comments and Suggestions for Authors

The manuscript vaccines-3092161 reports interesting results scientifically important. Based on my comments below, I recommend the publication after minor revisions:

1) In the UV-Vis absorption analysis some of the data are higher than the linearity of the Beer-Lambert law. Does it decrease the reliability of the raw data? Please, explain it in the manuscript.

2) The authors applied statistical analysis for some data. However, what is the reason that in some cases it was not considered? As an example, reactivity of inhibitory mAb CyP1.9 and isotype control, PfCyRPA cVLP, and Fragment A cVLP in the design, production, and immunizations…

3) What are the corresponding standard deviation values for Ka, Kd, and KD parameters?

4) In Figure 4b provides the units of the Y- and X-axis.

5) Please, provide the characteristics and preparation of the chip used to obtain the data reported in Figure 4b.

6) The authors provided the sensorgrams without the nonlinear regression curve fit used to determine the constants. Please, provide these data in the manuscript.

Author Response

Comments 1: In the UV-Vis absorption analysis some of the data are higher than the linearity of the Beer-Lambert law. Does it decrease the reliability of the raw data? Please, explain it in the manuscript.

Response 1: Thank you for pointing this out. We believe the comment is referring to the non-linearity exhibited in the Growth-inhibition activity assays in figure 2H and 3E. In these plots, the y-axis does not depict the absorbance values, but the growth-inhibition activity values (in percentage) calculated from the absorbance values. Thus, this non-linearity is not due to saturation of the absorbance in the assay, but to the number of viable parasites at high concentrations of inhibitory antibodies being very low and giving a signal close to that of the negative control wells containing no parasites (red blood cells only). I.e., saturation of the assay at 100 % growth inhibition actually means no signal as there are no viable parasites. We believe this is shown in the manuscript by the labels of the y-axis of the plots and the formula used to calculate the growth inhibition % is explained in the methods section under “growth inhibition acitivity”, starting from line 270. Please, do comment if this is not the issue you were referring to, or if you believe further elaboration in the manuscript is needed.

Comments 2: The authors applied statistical analysis for some data. However, what is the reason that in some cases it was not considered? As an example, reactivity of inhibitory mAb CyP1.9 and isotype control, PfCyRPA cVLP, and Fragment A cVLP in the design, production, and immunizations…

Response 2: Thanks for the question. We have now added a Kruskall-Wallis test with Dunn’s post test when comparing PfCyRPA specific response and total IgG production in figures 2 (page 9) and 3 (page 12) and added it to the figure legends. Further, the methods section has been elaborated about the tests, page 6, line 289-292.

An elaboration: There are multiple cases and different explanations to each case.

Case 1: Comparing data based on purified IgG from pooled serum (Figures 2g, 2h, 3d, 3e, 3f). In these cases, even if technical replicates were possible (as for the growth inhibition activity assays), we only had 1 biological replicate from each group as all the individual serum sample in each group were pooled. Statistical analysis was therefore not possible.

Case 2: Non-serum ELISAs (Figures 1a, 1b, 1d, 1e, 2c). For binding to peptides derived from the phage display, we were only interested in validating the binding to the target mAb. The phages were kept at a specific dilution and samples run in duplicates. This was not sufficient to run statistics, but we believed it was sufficient for the screening to find possible binders. For testing mAbs on our vaccine constructs, the reported data is area under the curve for a single titration and therefore insufficient for statistical analysis.

Case 3: Serum ELISAs for measuring of PfCyRPA specific response and serum IgG quantification (Figures 2e, 2f, 3b, 3c, 4a). We were interested in two approaches for increasing the neutralizing antibody response. The first one was coupling PfCyRPA to a cVLP and compare its efficacy as immunogen with that of soluble PfCyRPA. For all of these pairwise comparisons, we used a two-tailed Mann-Whitney test. The second one was multiple approaches to truncate PfCyRPA when displayed on a cVLP. Here, we have now performed Kruskall-Wallis test with Dunn’s post test to compare the various immunofocusing constructs and added the results in figures 2 and 3.     

Comments 3: What are the corresponding standard deviation values for Ka, Kd, and KD parameters?

Response 3: Thanks for taking this up, and we agree that these values would be beneficial. However, the ATTANA system and the analysis software Trace Drawer used to calculate the parameters does not report the standard deviations, and hence we are not able to provide them.

Comments 4: In Figure 4b provides the units of the Y- and X-axis.

Response 4: We acknowledge that the axis labeling was missing. The figure has been updated.

Comments 5: Please, provide the characteristics and preparation of the chip used to obtain the data reported in Figure 4b.

Response 5: To address this comment, the preparation of the chip described in the methods section has been elaborated (manuscript page 5 lines 253-264). Further, the figure legend to figure 4b has been elaborated. We have added most of the details to the methods section in order to keep the figure legend succinct.

Comments 6: The authors provided the sensorgrams without the nonlinear regression curve fit used to determine the constants. Please, provide these data in the manuscript.

Response 6: Thanks for your comment. We have indeed not made it explicit how the nonlinear regression curve is displayed and we have now added this information to the figure caption (figure 4). For each IgG concentration tested, curves representing the experimental data are in different colors (depicted in the figure legend), and the curve of the fitted model is superimposed in black.

Reviewer 3 Report

Comments and Suggestions for Authors

The authors have submitted the manuscript titled "Leveraging Immunofocusing and Virus-Like Particle Display to Enhance Antibody Responses to the Malaria Blood-Stage Invasion Complex Antigen PfCyRPA". In this, the authors used Immunofocusing and capsid Virus Like Particles (cVLPs) as strategies to increase the neutralizing antibody titers to Malaria vaccine. 

I have elaborated my comments on the study below:

1. The protein expression and purification in the methods section needs to be elaborated with all the details added including Expi293 cell culturing, cell-density used for protein expression.

2. In the mice immunization section in methods, the authors only mention that the vaccines were formulated in 50% volume AddaVax. What is the concentration of AddaVax per dose? What was the formulation vehicle?

3. The authors need to elaborate on the groups of mice used in the study in a tabular format including how many mice were used in each group (with appropriate male/female distribution used) and what each group was immunized with.

4. The authors only compared the results from the ice serum samples at day 27 and day 42. They do not have a baseline, nor a negative control group of mice (either untreated mice or mice immunized with PBS only). This must be added in for appropriate and fair comparisons.

Comments on the Quality of English Language

The manuscript has a few typos that need to be corrected. Other than that there are no issues with English Language.

Author Response

Comments 1: The protein expression and purification in the methods section needs to be elaborated with all the details added including Expi293 cell culturing, cell-density used for protein expression.

Response 1: We agree that this part of the methods section needed additional information. The Expi293 protein expression method has been elaborated in Materials and Methods section page 3, line 127-133.

Comments 2: In the mice immunization section in methods, the authors only mention that the vaccines were formulated in 50% volume AddaVax. What is the concentration of AddaVax per dose? What was the formulation vehicle?

Response 2: We would glady give the concentration, if only it was given by the manufacturer.Addavax is simply stated as an oil-in-water emulsion. Thus, 50% volume AddaVax is the only way we can describe the amount of Addavax present in each vaccine dose.. The formulation has been clarified in “Methods” under “Mice immunizations”, stating that the vaccines are diluted in PBS prior to mixing with equal volume of AddaVax. The methods have, however, been elaborated, page 4 line 190-196.

Comments 3: The authors need to elaborate on the groups of mice used in the study in a tabular format including how many mice were used in each group (with appropriate male/female distribution used) and what each group was immunized with.

Response 3: We acknowledge the need of an overview of the number of vaccinated mice and their sex. A supplementary table S1 has been added. Further, the number of mice per group has been added in figure legends for figures 2 and 3, as well as in the methods section (page 4 line 190-192).

Comments 4: The authors only compared the results from the ice serum samples at day 27 and day 42. They do not have a baseline, nor a negative control group of mice (either untreated mice or mice immunized with PBS only). This must be added in for appropriate and fair comparisons.

Response 4: Thanks for taking up this important point. For all the immunofocusing constructs, we believe the appropriate negative control is the SpyCatcher cVLP with no coupled antigen as this accounts for the baseline immune response to the cVLP vector and the adjuvant used. Full-length PfCyRPA displayed on cVLP is used as positive control.

To investigate whether displaying full-length PfCyRPA on a cVLP is better than soluble protein alone, we believe that soluble PfCyRPA is the appropriate control, as we are only interested in the difference between the two modes of display (soluble vs. cVLP). Further, it is our opinion that a PBS vaccinated mice group would have been the appropriate negative control if we were comparing vaccination with soluble PfCyRPA with vaccination with other protein antigens (no cVLPs). As this was not the case, we elected not to include a PBS vaccinated group.

Round 2

Reviewer 3 Report

Comments and Suggestions for Authors

In the revised manuscript, the authors have addressed most of my comments except the need for a negative control group. While I agree with not including the PBS-immunized mice as negative controls, I feel it is essential to have the baseline or pre-immunization responses of the mice as a comparison. The authors have stated that they used SpyCatcher cVLP with no coupled antigen as the negative control. However, this is not the appropriate negative control, this can be a no-antigen group, but since they are introducing cVLP, there might be an immune response. To truly understand if the cVLP gives a background immune response and the significance of the response generated, the authors need to include the baseline results.

Author Response

Comments 1: In the revised manuscript, the authors have addressed most of my comments except the need for a negative control group. While I agree with not including the PBS-immunized mice as negative controls, I feel it is essential to have the baseline or pre-immunization responses of the mice as a comparison. The authors have stated that they used SpyCatcher cVLP with no coupled antigen as the negative control. However, this is not the appropriate negative control, this can be a no-antigen group, but since they are introducing cVLP, there might be an immune response. To truly understand if the cVLP gives a background immune response and the significance of the response generated, the authors need to include the baseline results.

Response 1: We believe the best way to account for the immune response against the cVLP is to use the SpyCatcher cVLP as a baseline. In this way, for the immunofocusing constructs displayed on a cVLP, full-length PfCyRPA acts as a positive control and SpyCatcher cVLP as a negative control. We believe we thereby have accouted for the SpyCatcher cVLP specific response.

However, we agree that the cVLP alone will generate a response and that it is beneficial to know the baseline of PfCyRPA reactivity and serum IgG concentration. We have therefore included ELISA data of these metrics in the supplementary information, now supplementary figure S3.